# Mouse Model of Nitrogen Mustard Ocular Surface Injury Characterization and Sphingolipid Signaling

**DOI:** 10.3390/ijms25020742

**Published:** 2024-01-06

**Authors:** Sandip K. Basu, Amanda Prislovsky, Nataliya Lenchik, Daniel J. Stephenson, Rajesh Agarwal, Charles E. Chalfant, Nawajes Mandal

**Affiliations:** 1Department of Ophthalmology, The University of Health Science Centre, Memphis, TN 38163, USA; sbasu8@uthsc.edu (S.K.B.); aprislo1@uthsc.edu (A.P.); nlenchick@uthsc.edu (N.L.); 2Memphis VA Medical Center, Memphis, TN 38104, USA; 3Departments of Medicine and Cell Biology, University of Virginia School of Medicine, Charlottesville, VA 22903, USA; stephensondj@alumni.vcu.edu (D.J.S.); krj2sf@uvahealth.org (C.E.C.); 4Department of Pharmaceutical Sciences, University of Colorado Anschutz Medical Campus, Aurora, CO 80045, USA; rajesh.agarwal@cuanschutz.edu; 5Research Service, Richmond Veterans Administration Medical Center, Richmond, VA 23298, USA; 6Department of Anatomy and Neurobiology, The University of Health Science Centre, Memphis, TN 38163, USA

**Keywords:** nitrogen mustard (NM), NM eye injury, mouse model, vesicating injury, ocular surface injury, inflammation, sphingolipids, visual function

## Abstract

Vesicating chemicals like sulfur mustard (SM) or nitrogen mustard (NM) can cause devastating damage to the eyes, skin, and lungs. Eyes, being the most sensitive, have complicated pathologies that can manifest immediately after exposure (acute) and last for years (chronic). No FDA-approved drug is available to be used as medical counter measures (MCMs) against such injuries. Understanding the pathological mechanisms in acute and chronic response of the eye is essential for developing effective MCMs. Here, we report the clinical and histopathological characterization of a mouse model of NM-induced ocular surface injury (entire surface) developed by treating the eye with 2% (*w*/*v*) NM solution for 5 min. Unlike the existing models of specific injury, our model showed severe ocular inflammation, including the eyelids, structural deformity of the corneal epithelium and stroma, and diminished visual and retinal functions. We also observed alterations of the inflammatory markers and their expression at different phases of the injury, along with an activation of acidic sphingomyelinase (aSMase), causing an increase in bioactive sphingolipid ceramide and a reduction in sphingomyelin levels. This novel ocular surface mouse model recapitulated the injuries reported in human, rabbit, and murine SM or NM injury models. NM exposure of the entire ocular surface in mice, which is similar to accidental or deliberate exposure in humans, showed severe ocular inflammation and caused irreversible alterations to the corneal structure and significant vision loss. It also showed an intricate interplay between inflammatory markers over the injury period and alteration in sphingolipid homeostasis in the early acute phase.

## 1. Introduction

Vesicating or blister-forming chemicals have been used as chemical warfare agents since their first use in 1917 during World War I until recently in Syria in 2016 [1,2]. The most common agents among them are sulfur mustard (SM), nitrogen mustard (NM), arsenical vesicant lewisite (LEW), and nettle agent phosgene oxime (CX) [3]. The primary modes of exposure to these chemicals are contact, inhalation, and ingestion, and they cause moderate to severe injuries to the skin, lungs, and eye, causing blisters, bronchospasm, pulmonary edema, bronchitis, immune suppression, and ocular and dermal burns [4,5,6]. No therapeutic options known as medical counter measures (MCMs) are available to treat vesicant injuries, and no FDA-approved drug is available as an MCM [7,8]. So, there is a dire need to understand the pathophysiology and underlying mechanisms of vesicant injuries to develop effective therapies and identify targets for developing potential MCMs. The eyes are particularly vulnerable among the primary target organs of vesicating agents, with an almost 10-fold higher sensitivity to these agents than the skin [9,10]. The ocular injury caused by these agents manifests quicker than skin injury and requires less exposure time and a lower concentration of the toxicant [7].

As per the existing knowledge, the ocular pathology of vesicant injury is highly complex and the mechanisms are poorly understood. Several studies in ex vivo rabbit corneas and in vivo rabbit and mouse models reported that vesicant agents such as SM, lewisite, and NM cause a biphasic injury to the eye [11,12,13,14,15,16,17,18,19,20,21]. Studies from primary human corneal epithelial cells, rabbit corneal organ culture, and ex vivo rabbit corneas have shown that immediately after NM injury, in the acute phase, there is a severe inflammatory response that resolves as the acute phase of the injury subsides. A delayed injury phase ensues, characterized by chronic and persistent inflammation leading to multiple pathological symptoms like edema, corneal erosions, and scar formation [11]. Similar studies with the exposure of LEW vapor showed increased corneal thickness and induced epithelial degradation and epithelial–stromal separation [11,22]. Further studies are therefore necessary to understand the mechanisms that govern the initial acute phase and how that transforms into a chronic disease. Developing animal models with a variety of exposure and clinical characterization for short-term and long-term effects on the entire visual system and investigating novel biochemical and molecular entities in the mechanisms of the injury and pathology will help to narrow down some potential targets helpful in developing an MCM. In this study, we developed an ocular surface injury model of mice by exposing them to NM and determined its pathological association with lipid mediators.

While undertaking this study, we found that other groups reported the development of mouse models of NM exposure in the last 2 years [23,24,25,26]. Clinical and histopathological characterization of these models suggested that they can recapitulate the injury reported in human and in vivo rabbit models. Some of the studies focused on the implication of cellular senescence [25] or the potential of mesenchymal stem cell (MSC) therapy following NM-induced keratopathy [26]. Some of these studies developed the model by placing filter paper soaked in an NM solution at the central cornea [23,24]. We postulated that SM or NM ocular exposure in humans, both accidental or in chemical warfare, will not be restricted to the central cornea but will affect the entire ocular surface; therefore, we developed a mouse model of NM-exposed ocular surface injury (including cornea conjunctiva and sclera) to mimic human exposures more closely.

Sphingolipids are a subclass of cellular lipids that, in addition to providing structural support, play an active role in cellular signaling and regulating multiple physiological processes [27,28,29]. Bioactive sphingolipids (SPLs) have been shown to play important roles in ocular diseases, including corneal injury, neovascularization, inflammation, and fibrosis [30,31,32,33,34,35]. Activation of sphingomyelinase enzyme (SMase) and generation of bioactive SPLs such as ceramide (Cer), ceramide 1-phosphate (C1P), sphingosine, and sphingosine 1-phosphate (S1P) is considered to initiate and maintain acute and chronic inflammation [36,37]. Since inflammation appears to be a key component for acute and chronic vesicating injury pathology, it is likely that bioactive sphingolipid signaling will play an important role in vesicating injury. However, lipid signaling is largely an unexplored area in the pathophysiological mechanisms of SM- or NM-mediated ocular injury. Only one study so far has reported that alteration of ceramide and sphingomyelins occurs in an ex vivo model of NM injury of rabbit cornea [38]. Thus, the role of sphingolipid signaling in the mechanism of vesicating ocular injury requires further investigation.

In this study, we report the development of a novel mouse model of NM-induced ocular surface injury and detailed clinical and histopathological characterization of the model both at the acute and chronic phases of the injury. We posit that this model will mimic human injury more closely and provide an important tool for studying the effect of potential MCM candidates. We also report characterization of the ocular surface inflammatory markers and signaling sphingolipids following injury and their effect on the visual and retinal functions. This study provides novel insights into the manifestation of inflammation over the injury period and the interplay between inflammatory and sphingolipid signaling.

## 2. Results

### 2.1. Developing a Mouse Model of NM-Induced Ocular Surface Injury

As described in the Section 4, we developed a mouse model of ocular surface injury. For clinical evaluation, we captured external images of the eyes with a digital camera and scored them based on visible inflammation by the closure of the eyes. The saline-treated (uninjured control) eyes do not show any visible signs of swelling of the eyelids or eye closure between 1 day post-injury (DPI) and 35 DPI (Figure 1A,I). However, in the NM-injured mice, visible signs of eye injury resulting in almost complete closure of the eyes by the swollen eyelids started appearing by the next day of injury (Figure 1B). We observed severe inflammation with white discharge on 3 DPI (Figure 1C), with the eyes remaining severely inflamed and swollen till 7 DPI (Figure 1D,E). Between 11 and 14 DPI, we observed moderate improvement of inflammation and clearing of the ocular surface (Figure 1F,G), with almost complete resolution of inflammation by 21 DPI. By 35 DPI, the NM-treated eyes were presented with visible opacity (cloudy), whitening of the surrounding eyelids, and loss of all eyelashes (Figure 1H). We scored the images based on a clinical scale of 1 to 4 as described in the Section 4 (where 1 is uninflamed and fully open eyes, 4 is inflamed and more than 75% closed eyes). While the saline-treated eyes consistently scored around 1, most NM-injured eyes ranged between 2.5 to 3.5 till 14 DPI and improved to ~1.5 in the next three weeks (Figure 1J). Based on this characterization, we divided the pathology into different stages: early acute (1–6 DPI), late acute (7–14 DPI), and chronic phases (21–35+ DPI). Taken together, the data suggested that we have successfully developed a mouse model NM-induced ocular surface injury that manifests similar eyelids and conjunctival inflammation and restricted eye openings as reported in in vivo rabbit models [39].

### 2.2. Histopathological Characterization of NM-Injured Mouse Corneas

Our histological evaluation of the mouse ocular surface revealed a pathology that changes over the course of the injury phases. Compared to the saline-treated mice cornea (Figure 2A,B), corneal edema with loss of stromal collagen organization was observed by 3 DPI, along with stromal swelling and corneal erosion (Figure 2C,D; arrows). By 7 DPI, epithelial detachment was observed on the cornea (Figure 2E,F; arrows). In most mice, the corneal epithelium was left with a single layer of epithelial cells (Figure 2E,F, arrows), and the corneal stroma was swollen, showing a complete dysmorphic collagen structure (Figure 2E,F; arrows). The corneal epithelia appeared to be healed by 35 DPI (Figure 2G,H, arrows); however, structural abnormality is still observed in corneal stromal collagen organization, with the collagen being either found to be compacted towards the endothelium (Figure 2H; arrows) or throughout the cornea. Thus, the histopathological effect of NM on the mouse ocular surface seems to manifest a similar pathology to that observed in rabbit cornea following NM or lewisite exposure [39].

### 2.3. Characterization of Inflammatory Markers following NM Injury

The quantitative (pg/mL) expression level of inflammatory marker proteins was assayed from saline- and NM-treated mice at the early acute phase (3 DPI), the start of the late acute phase (7 DPI), and in the chronic phase (35 DPI) (Figure 3). The data suggest that the inflammatory markers on the ocular surface can be broadly classified into three groups. The first group’s expression peaks at the early acute phase and slowly reduces by the start of the late acute phase. Their expression goes down by the chronic phase to levels comparable to the saline-treated mice. This group includes Interferon (IFN) gamma, Interleukin-1 (IL-1) alpha and beta, Monocyte-chemotactic protein-3 (MCP-3), and Tumor necrosis factor (TNF) alpha (Figure 3A–E). The second group’s expression level peaked either at the early acute phase and maintained by the late acute phase or increased by the early acute phase but peaked by the late acute phase. In either case, the expression level goes down by the chronic phase. This group includes IL-2, Macrophage inflammatory protein-2 (MIP-2) alpha, and Vascular endothelial growth factor (VEGF) alpha (Figure 3F–H). The third group contains the least markers among the tested ones and is characterized by persistent high expression levels well into the chronic phase. This group contains Eotaxin for the corneal surface (Figure 3I).

### 2.4. Effect of NM-Induced Ocular Surface Injury on the Visual Functions

We tested the vision of the mice before injury (Pre) and again at 14 DPI (Post-2wk) and 35 DPI (Post-5wk) by optokinetic nystagmus (OKN). Both the visual acuity and the contrast sensitivity of the NM-treated mice were severely affected (decrease in visual acuity and increase in contrast sensitivity), leaving them almost completely blind (close to 100% contrast sensitivity) as observed during the acute phase of the injury (14 DPI) as well as in the chronic phase of the injury (35 DPI) (Figure 4A,B).

The visual or retinal functions were tested by electroretinography (ERG) and were not completely diminished in the NM-injured mice. However, both the scotopic negative A-(−A: evaluate rod function) and B-waves (evaluate the functions of secondary neurons) were significantly reduced in the NM-treated mice compared to the saline-treated controls at 14 and 35 DPI (Figure 5A,B).

### 2.5. Characterization of Sphingolipid Signaling in NM-Induced Ocular Surface Injury

The enzyme activity of both aSMase (acidic) and nSMases (neutral) from mouse corneal surfaces at 3, 7, and 35 DPI were depicted in Figure 6. We observed that compared to saline-treated mice, the activity of aSMase significantly increased in the early acute phase (3 DPI) in the NM-injured mice. The activity decreased by the late acute phase (7 DPI) and was maintained at a low activity state into the chronic phase (35 DPI) (Figure 6A). Interestingly, nSMase activity was not significantly changed in both early and late acute phases compared to the saline-treated mice; however, it was significantly increased in the chronic phase of the injury (Figure 6B).

In order to determine the levels of different sphingolipids, we next conducted sphingolipid profiling of the ocular surface from NM-treated mice along with saline-treated controls, and, as a pilot study, we analyzed the samples from only one time point, 3 DPI, where at the early acute phase we noticed an increase in aSMase (Figure 6A). Consistent with the increase in aSMase activity at 3 DPI, we observed a decrease in the Sphingomyelin (SM) from 83% to 74% with concomitant increases in ceramide (6% to 8%) and hexosylceramide (HexCer: 11% to 18%) (Figure 7A,B).

Analysis of individual species of these sphingolipids showed a similar pattern. Multiple species of ceramide (Figure 8A) and HexCer (Figure 8B) showed a significant increase in their levels in the treated mice, with significant reductions in multiple SM species in the NM-treated mice (Figure 8C). The only exception was observed in C26:0 in the treated ocular surface, where with a highly significant decrease in SM, we observed a highly significant increase in HexCer level but observed a decrease in the corresponding ceramide levels (Figure 8A–C). We analyzed one dihydro (DH) species, C16DH, an intermediate of de novo Cer biosynthesis, and found its level increased significantly in ceramide but reduced significantly in SM, like any other non-dihydro species (Figure 8A,C). Analysis of the other bioactive sphingolipids showed a significant increase in sphingosine levels in the ocular surface following NM exposure. We also observed a decrease in dihydro-sphingosine along with the levels of 22:0 C1P and 24:1 C1P following NM treatment (Figure 8D).

## 3. Discussion

In this study, we reported the development of a novel mouse model of NM-induced ocular surface injury and conducted detailed clinical and histopathological characterization of the model. SM, NM, LEW, and CX are classified into blister-forming or vesicating chemicals designated as toxins of concern for public health and safety [7]. SM and NM are highly reactive alkylating agents that covalently modify all major cellular biomolecules, such as DNA, proteins, and lipids, and cause both acute and chronic injuries to the affected organs [7,8,21]. SM and NM can induce severe ocular injury, and the features of this injury have been well documented in humans (SM only) [40,41] and animal models (rabbits and rodents for SM and rabbits and mice for NM) [11,12,13,14,15,16,17,18,19,20,21]. NM is an analog of SM, and being commercially available, NM can serve as an excellent surrogate to study the ocular injuries resulting from SM exposure. However, damage to the eye of victims of either chemical warfare or accidental exposure will never be restricted to the center of the cornea, but the whole ocular surface will be exposed to the toxin. To recapitulate this condition experienced in a real-life situation, we developed a model of whole ocular surface injury, including the cornea, conjunctiva, and sclera, which is different from other recently developed mouse models of NM-mediated corneal injury [23,24,25,26], and this is the first report on characterizing NM-induced mouse ocular surface injury. We aimed to develop a model of moderate injury that would be useful to test MCM candidates. Based on a series of pilot assessments, we chose an exposure to 2% (*w*/*v*) NM solution directly to the ocular surface for 5 min. Clinical characterization revealed severe eye and eyelid inflammation leading to the closure of the eye, starting from 1 DPI with white discharge observed by 3 DPI and severely swollen eyes until 7 DPI (Figure 1B–E). The swelling of the eyes was improved with the discharge getting cleared by 14 DPI (Figure 1G); however, even at 35 DPI, the eyes were cloudy compared to the saline-treated eyes with loss of eyelashes (Figure 1 H,I). We observed this NM-induced entire ocular surface injury model demonstrate a longitudinal pathology with an early acute phase (1–6 DPI) that progressed into a late acute phase (7–14 DPI) and ultimately to a chronic phase (21–35+ DPI). Identifying these sub-phases can help test and fine-tune the delivery window of drugs developed as MCMs against such injury.

Another unique feature of our study is that we evaluated the visual and retinal functions of the NM-exposed mice and their pathological association with sphingolipid metabolism and signaling. Along with the irreversible structural change of the corneal stroma observed at 35 DPI (Figure 2G,H), we observed that this caused a significant reduction in the visual and retina functions in mice (Figure 4A,B and Figure 5A,B). Both the visual acuity and contrast sensitivity were diminished to the level of near blindness at 14 DPI. The compromised retinal function (Figure 5A,B) could be either due to reduced light transmission through the cornea because of opacity or the direct effect of the NM solution on the retina by spreading from the ocular surface into the posterior part, or due to a combination of both of them, which needs further investigation. We observed an increase of aSMase activity consistent with an increase in Cer and HexCers and a decrease in SM in the early acute phase (Figure 6, Figure 7 and Figure 8), which clearly indicates the generation of Cer from SM by aSMase activation and could be a key early event of NM-induced ocular pathology and a potential target for MCM development.

To compare the pathology of our model in three identified distinct phases, in the early acute phase (1–6 DPI), we observed severe inflammation of the eye, conjunctiva, and eyelids that restricted the opening of the eye along with whitish discharge covering the eye. This unique observation has not been reported so far and could be specific for a model of ocular surface injury. Histologically, we observed corneal edema with loss of stromal organization and swelling. Mouse models of NM injury from other research groups also reported corneal epithelial injury, loss of endothelial cells, and stromal edema within 3 DPI [23,24]. In the late acute phase, our data compares well with the published models, including corneal opacity, epithelial detachment, and edema. However, our model differs from the published models in the completely disorganized stroma with dysmorphic collagen structure in the late acute phase. In the chronic phase, we detected a cloudy and opaque cornea as reported in the published mouse models of NM injury, but the complete loss of eyelashes along with compacted collagen towards the endothelium is something that we found to be unique in our model. This disorganization of the collagen observed in the corneal stroma affected the corneal transparency and led to the compromised visual functions observed in our model. In conclusion, though we exposed the entire ocular surface to NM, the corneal pathology of our model compares very well with only cornea exposure models of mice and rabbits. Additionally, our methods resulted in significant pathology of the eyelids, affecting meibomian and lacrimal glands and causing dry eyes, which is under detailed investigation.

Inflammation is the key pathological component of vesicating ocular injuries [42,43], which is recapitulated in our model. We subjected NM-exposed ocular surface tissues to a quantitative measurement (pg/mL) of >40 inflammatory markers using a multiplex assay platform at different phases of the injury. We observed distinct expression patterns, elevation, or activation patterns of those with the three phases of injury and healing. Inflammatory signaling proteins that elevated in the early acute phase but subsided in the late acute and chronic phases include IFN gamma, IL-1 alpha and beta, MCP-3, and TNF alpha (Figure 3A–E). Proteins that stayed elevated in the acute phases but maintained reduced expression levels in the chronic phase include IL-2, MIP-2 alpha, and VEGF alpha (Figure 3F–H). We found that Eotaxin elevation was maintained through all the phases (Figure 3I). As expected, this pattern indicates a complex interplay between the inflammatory proteins. The expression of the majority of those tends to reduce as the injury enters from the resolving phase, where the resolution of inflammation is necessary for proper tissue repair [44,45]. Eotaxin may represent a marker for chronic inflammation in the ocular surface since higher expression of Eotaxin has been implicated in chronic asthma and chronic sinusitis [46,47].

Lipid signaling in vesicating ocular injury or skin, or lung injury has not been explored in detail. However, the inflammatory pathology as the key to any vesicating injury suggests their integral association with lipid signaling. Bioactive lipids such as eicosanoids, sphingolipids, specialized pro-resolving mediators (SPM), and endocannabinoids are classic cellular and tissue mediators that are involved not only in initiating and potentiating inflammation but also in the ultimate cessation or resolution of the inflammatory responses [36,48,49,50]. With our eventual goal of understanding lipid signaling in the pathological mechanisms of vesicating injuries, in this maiden step, we determined the association of sphingolipids in an acute stage of NM-exposed mice eyes (3 DPI). Consistent with a previous observation of NM-exposed ex vivo rabbit cornea, our results showed activation of acidic SMase in the acute phase but not neutral SMase (Figure 6A,B) [38]. SMase hydrolyzes SM to produce phosphorylcholine and Cer, and this is a key step in sphingolipid signaling. Cer can be the precursor for other bioactive sphingolipids such as sphingosine, S1P, and C1P. They regulate apoptosis, cell growth, inflammation, angiogenesis, intracellular trafficking, and other processes [36,50,51,52,53,54]. In general, they all have proinflammatory effects, with both pro- and anti-apoptotic roles for some of them. Shorter-chain-containing Cer are known to be more bioactive than the longer-chain species. The interconvertibility of these molecules and their biological effects often oppose each other, giving rise to the concept of a ‘sphingolipid rheostat,‘ which maintains the balance of these molecules and thus cellular homeostasis [50,55]. Cytokines and activated tumor necrosis factor receptors (TNFR) have been shown to activate aSMases, which, in turn, convert membrane-bound SM to Cer during the early stages of injury [56,57,58]. Cer then serves as an essential facilitator of downstream signaling and proinflammatory gene transcription by NF-κB [57,58,59]. Our observation from in vivo models of aSMase activation consistent with an increase in Cer in the early acute phase not only supports this notion but also associates sphingolipid signaling with NM ocular injury and thus forms the basis for further characterization in all different phases of injury and tissue healing to understand the sphingolipid mediation in the molecular pathology of vesicating ocular injury.

In conclusion, we developed and characterized the mouse model of NM-induced ocular surface injury, which manifests ocular surface inflammation and pathology similar to those observed in mouse and rabbit models. We also provide the quantitative assessment of the inflammatory markers across the injury phases, and the results indicated a complex interaction between the players that required further detailed investigation. Our data also suggest that sphingolipid homeostasis of the ocular surface is affected by NM. Since the connection between bioactive sphingolipids and inflammation is well established, it also indicates the presence of another crosstalk between two different classes of macromolecules, proteins, and lipids in the manifestation of the injury.

## 4. Materials and Methods

### 4.1. Animal Care

Male and female C57/BL6 mice that were 10–12 weeks old were utilized in this study. The mice were maintained in dim (5–10 lux) cyclic light (12 h ON/OFF) from birth. All the animals were born and raised in the University of Tennessee Health Science Center (UTHSC) vivarium, following its guidelines of animal housing. All procedures were performed according to the Association for Research in Vision and Ophthalmology Statement for the Use of Animals in Ophthalmic and Vision Research. The procedures also followed the UTHSC Guidelines for Animals in Research and were reviewed and approved by the UTHSC Institutional Animal Care and Use Committee (IACUC) (IACUC Protocol # 23-0425).

### 4.2. Ocular Surface Injury by NM and Clinical Characterization of NM-Induced Injury

The ocular surface injury was performed in mice anesthetized with an intraperitoneal (IP) injection of ketamine (100 mg/kg body weight) and xylazine (5 mg/kg body weight). The ocular surface injury was achieved by adding 10 µL of 2% NM solution (*w*/*v* in 0.9% sterile saline) in each eye, encompassing the entire ocular surface of the anesthetized mouse for 5 min, followed by washing the eyes with 0.9% saline solution for 20 s. Similar-age-matched littermates treated with 10 µL 0.9% saline in each eye for 5 min served as uninjured controls. Following treatment, the mice were kept on a heating pad until they regained consciousness and were returned to their original housing. Both injured and uninjured control groups were used for the characterization of NM-induced ocular surface injury.

For clinical evaluation and scoring, mice ocular surface was imaged at different DPI, starting from 1 DPI to 14 DPI, representing the acute phase of the injury and again at 35 DPI, representing the chronic phase of the injury by digital camera. The images were scored by five independent evaluators for visible signs of inflammation and closure of the eye and eyelids. The scale used for the clinical score was 1 = 100% open; 2 = 75% open; 3 = 50% open; and 4 = <25% open. A total of six mice of both genders per treatment group (NM and saline) were used for both ocular surface injury and clinical evaluation.

### 4.3. Histopathological Characterization of NM-Induced Injury

The histopathological characterization was carried out at 3 DPI (early acute phase), 7 DPI (start of late acute phase), and 35 DPI (chronic phase). Mice from both uninjured and injured groups were euthanized at the specific DPI, and the eyes were enucleated. The eyes were fixed by Prefer fixative (Anatech Ltd., Battle Creek, MI, USA), processed for paraffin-embedded histology, and cut into 10-micron sections using a rotary microtome (Epredia HM 325, Fisher Scientific, Waltham, MA, USA) and stained by H&E following published protocols [60]. The sections were imaged using an Eclipse 80i fluorescence microscope (Nikon, Melville, NY, USA). Six mice of both genders were used for each treatment group (NM and saline), with one eye from each mouse being processed for histology.

### 4.4. Multiplex Analysis of the Inflammatory Markers of the Ocular Surface following NM Injury

To characterize the inflammatory markers at different phases of NM injury, mice from uninjured and injured groups were euthanized at 3, 7, and 35 DPI. The eyes were enucleated, and the ocular surface (including the cornea, conjunctiva, and sclera) was dissected. Total proteins were isolated from these tissues and quantified following published protocols [61]. Mouse ProcartaPlex # EPX 480-20834-901 Thermo Fisher Scientific, Waltham, MA, USA), a multiplex assay format, was used to quantify 43 inflammatory marker proteins quantitatively (pg/mL) from the ocular surface. The antibodies include, BAFF, betacellulin (BTC), ENA-78 (CXCL5), Eotaxin (CCL11), GM-CSF, GRO alpha (CXCL1), IFN alpha, IFN gamma, IL-1 beta, IL-2, IL-2R, IL-3, IL-4, IL-6, IL-7, IL-7R alpha, IL-10, IL-13, IL-15, IL-17A (CTLA-8), IL-18, IL-19, IL-22, IL-23, IL-25 (IL-17E), IL-27, IL-28, IL-31, IL-33, IL-33R (ST2), IL-alpha, LIF, Leptin, M-CSF, MCP-1 (CCL2), MCP-3 (CCL7), MIP-1 alpha (CCL3), MIP-1 beta (CCL4), MIP-2 alpha (CXCL2), RANKL, RANTES (CCL5), TNF Alpha, and VEGF-A. Thermo Fisher supplied all the detection and incubation components along with the custom panel. They also provided the standards and the sample to be used as a positive control. Negative controls were ocular surface tissue lysate without the magnetic beads and magnetic beads incubated in tissue lysis buffer. Following the manufacturer’s protocol, the magnetic beads containing 48 antibodies were added to a 96-well plate and then washed using a Hand-Held Magnetic Plate Washer (Invitrogen, Waltham, MA, USA; Catalog # EPX55555-000) incubated with 50 uL of protein samples, standards, or controls with continuous shaking overnight at 4 °C. The beads were then washed and incubated with detection antibody with shaking for 30 min. This was followed by 30 min with Streptavidin-PE conjugate at room temperature and protected from light. Then, the beads were washed and incubated for 5 min with 120 μL of reading buffer at room temperature. The plate was read on a Bio-Rad MagPix Multi Reader with the Bio-Plex Manager (Bio-Rad, Hercules, CA, USA) at the VA Medical Center (Memphis, TN, USA). The data were analyzed using Luminex’s ProcartaPlex analysis software (ProcartaPlex Analyst 1.0) available through Thermo Fisher Scientific (Waltham, MA, USA). Six mice of both genders were used for each treatment group (NM and saline), and one eye from each mouse was processed for multiplex analysis.

### 4.5. Analysis of Visual Function by Optokinetic Nystagmus (OKN) following NM Injury

The visual acuity and contrast sensitivity were measured by OKN using the OptoMotry system of Cerebral Mechanics (Lethbridge, AB, Canada). Both visual acuity and contrast sensitivity were measured once before injury and again at 14 DPI and at 35 DPI. The visual acuity was assessed at 100% contrast by varying the spatial frequency threshold while the contrast sensitivity was measured at a spatial frequency of 0.042 cycles per degree (c/d). Six mice of both genders were used for each treatment group (NM and saline). The visual acuity and contrast sensitivity data represented for each mouse were the average of the data from both the eyes of the individual mice.

### 4.6. Analysis of Retinal Function by Electroretinogram (ERG) following NM Injury

Retinal functions were analyzed by ERG once before injury and again at 14 DPI (end of late acute phase) and 35 DPI (chronic phase) to determine the effect of NM injury on retinal functions. Both scotopic and photopic flash ERGs were recorded using the Celaris ERG system (Diagnosys LLC, Lowell, MA, USA). Mice were dark-adapted overnight and were anesthetized under dim red light with ketamine (100 mg/kg body weight) and xylazine (5 mg/kg body weight) via IP injection. For dilation of the pupil, a drop of 1% (*w*/*v*) atropine and 1% (*w*/*v*) tropicamide (Akorn Inc., Lake Forest, IL, USA) were applied to the eye. For ERG measurement, designated electrodes were placed on each cornea, and the ERG was conducted using the TOUCH/TOUCH protocol developed by the manufacturer. For scotopic ERG, five flash stimuli were presented at flash intensities at 0.001, 0.01, 0.1, 1, and 10 cd.s/m^2^. The amplitude of the A-wave was measured from the pre-stimulus baseline to the A-wave trough, and the amplitude of B-wave was measured from the A-wave’s trough to the B-wave’s peak. Six mice of both genders were used for each treatment group (NM and saline). Both A- and B-waves of scotopic ERG for each mouse were the averages of the data from both eyes of the individual mouse.

### 4.7. Analysis of the Activity of the Sphingomyelinase Enzyme (SMase) following NM Injury

The activity of acidic and neutral sphingomyelinase (aSMase and nSMase, respectively) was measured from the ocular surface of saline-treated and NM-treated mice at different DPI using the Amplex^®^ Red Sphingomyelinase Assay kit (Invitrogen, Waltham, MA, USA) following a previously published protocol from our group [60]. In this method, SMase activity is measured indirectly in an enzyme-coupled assay using a microplate reader. The SMase present in the tissue hydrolyzes the sphingomyelin (supplied in the reaction mixture) to yield ceramide and phosphorylcholine. After the action of alkaline phosphatase, which hydrolyzes phosphorylcholine, choline is oxidized by choline oxidase to betaine and H_2_O_2_. Finally, H_2_O_2_, in the presence of horseradish peroxidase, reacts with Amplex^®^ Red reagent to generate the highly fluorescent product, which is measured at the absorption and emission maxima of ~571 nm and 585 nm, respectively. Thus, the SMase activity in represented using the relative fluorescence unit (RFU) with a higher RFU associated with higher SMase activity and a lower RFU with lower activity. This methodology can be used to continuously assay SMase enzymes with near-neutral pH optima (pH~7.4) for nSMase. Acidic SMase activity can be measured in two steps, where an aSMase reaction is performed at a lower pH (pH 5.0), followed by raising the pH to 7.0–8.0 to allow detection with the Amplex^®^ Red reagent. nSMase and aSMase RFUs were normalized with their protein content and used for statistical analysis. Six mice of both genders were used for each treatment group (NM and saline), and one eye from each mouse was processed for sphingomyelinase assay.

### 4.8. Sphingolipid Analysis of the Ocular Surface following NM Injury

In order to determine the changes in sphingolipids and also to understand sphingolipid-mediated signaling in ocular NM injury, the profile of major sphingolipids, including the known bioactive species, were analyzed from the ocular surface (including the cornea, conjunctiva, and sclera) from both saline- and NM-treated mice at the acute phase of the injury (3 DPI). Following euthanasia and enucleation, tissues were dissected and were snap frozen in liquid nitrogen and stored in a −80 °C freezer until further analysis. The sphingolipid analysis was performed following previously published protocols using UPLC ESI-MS/MS [62,63]. Briefly, a Shimadzu Nexera X2 LC-30AD with an Acentis Express C18 column (5 cm × 2.1 mm, 2.7 μm) was used to separate the sphingolipids at a flow rate of 0.5 mL/min at 60 °C. The column was equilibrated with Solvent A [methanol:water:formic acid (58:44:1, *v*/*v*/*v*) with 5 mM ammonium formate] for 5 min, followed by injecting 10 μL of the sample and eluting with 100% Solvent A for the first 0.5 min, transitioning to Solvent B [methanol:formic acid (99:1, *v*/*v*) with 5 mM ammonium formate] with a linear gradient to reach 100% Solvent B from 0.5 to 3.5 min. The sphingolipids were analyzed using an AB Sciex Triple Quad 5500 Mass Spectrometer and identified based on their retention time and *m*/*z* ratio. Semi-quantitative species determination was conducted by measuring the peak area of internal standards added to the samples [62]. Six mice of both genders were used for each treatment group (NM and saline), and one eye from each mouse was processed for sphingolipid analysis.

### 4.9. Statistical Analysis

All statistical analysis was performed using Student’s *t*-test and two-way ANOVA according to the experimental condition using GraphPad Prism 10 analysis software. Statistical significance was accepted at *p* < 0.05.

## Figures and Tables

**Figure 1 ijms-25-00742-f001:**
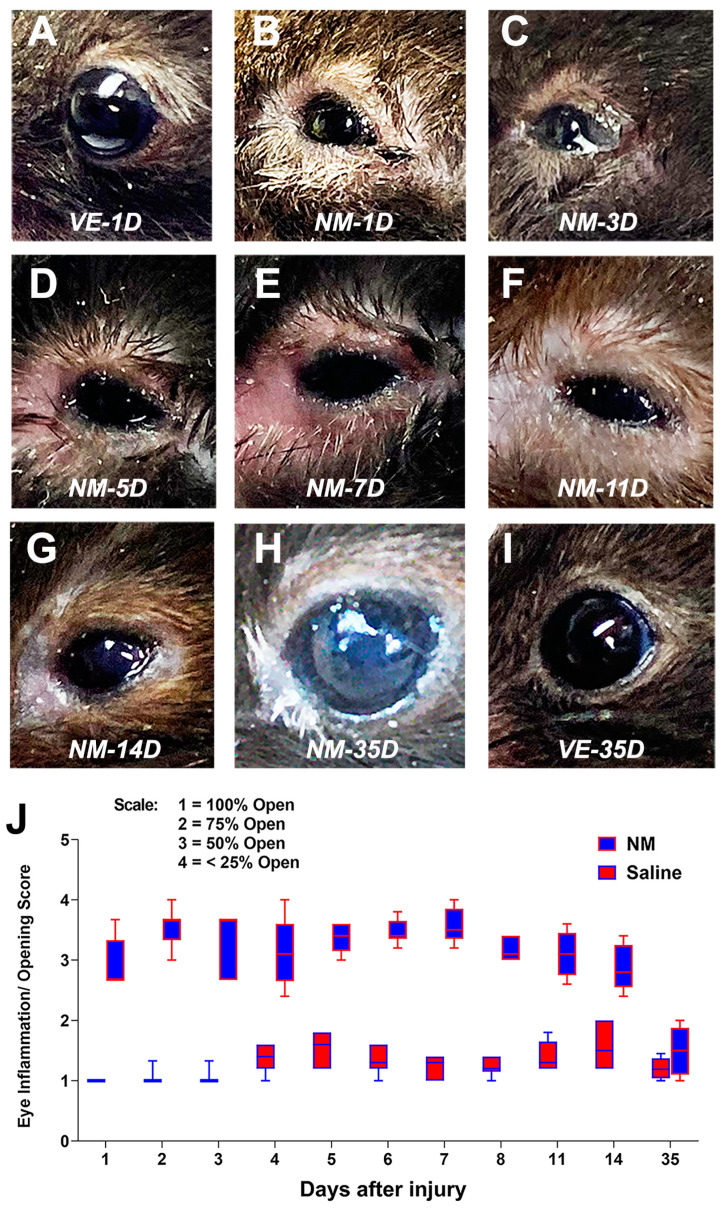
Clinical progression of ocular surface injury following NM exposure. Representative eye images of mice exposed to saline (vehicle) treatment (VE) at 1 DPI (**A**) and 35 DPI (**I**) and of mice exposed to 2% NM for 5 min (**B**–**H**) as described in the Section 4. Quantitative evaluation of the eye inflammation based on clinical scoring at different days post-injury of saline and NM-exposed mice (**J**). The data presented are mean ± SEM (*n* = 6). The images were captured with a digital camera, and the scale of each image is close but not the same to show an entire eye in a frame.

**Figure 2 ijms-25-00742-f002:**
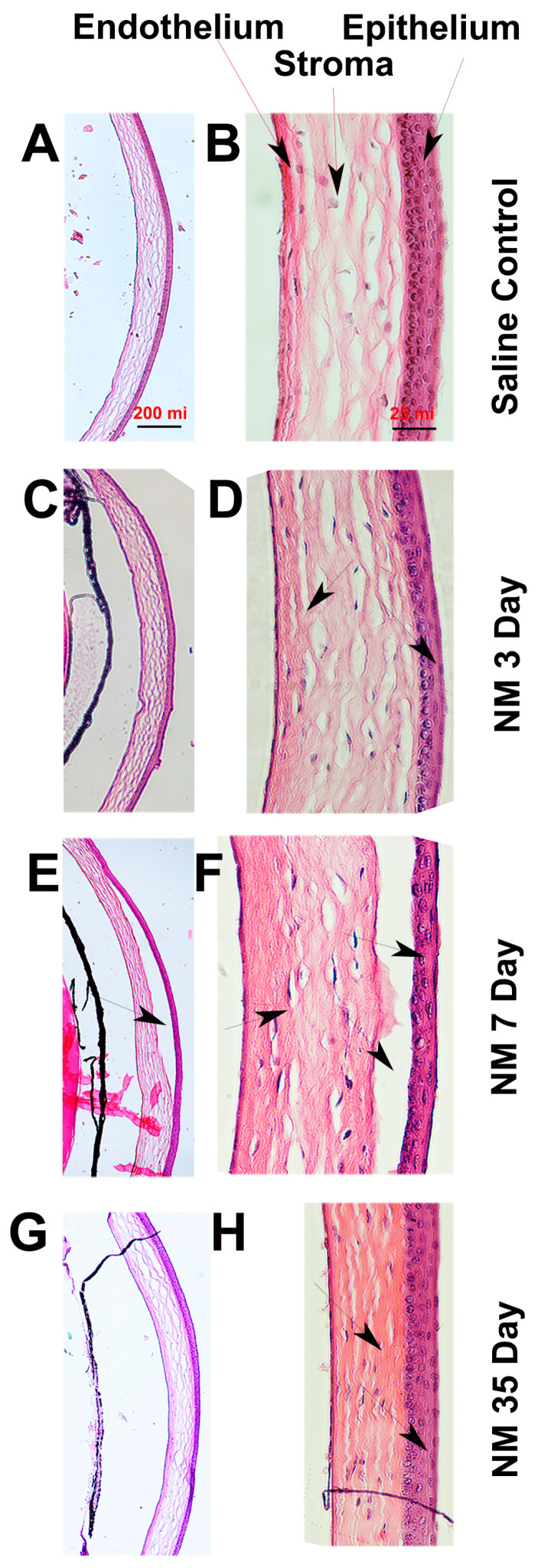
Histological evaluation of corneal structure following NM exposure. Representative images of hematoxylin and eosin (H&E) stained sections showing corneal structure from saline-treated (**A**,**B**) and NM-exposed (**C**–**H**) mice eyes. Images of the same eye for each time point was shown in 4× magnification ((**A**,**C**,**E**,**G**); scale bar: 200 micron) and 20× magnification ((**B**,**D**,**F**,**H**); scale bar: 25 micron). The arrows indicate the location of specific changes in the corneal structure as described in Section 2.

**Figure 3 ijms-25-00742-f003:**
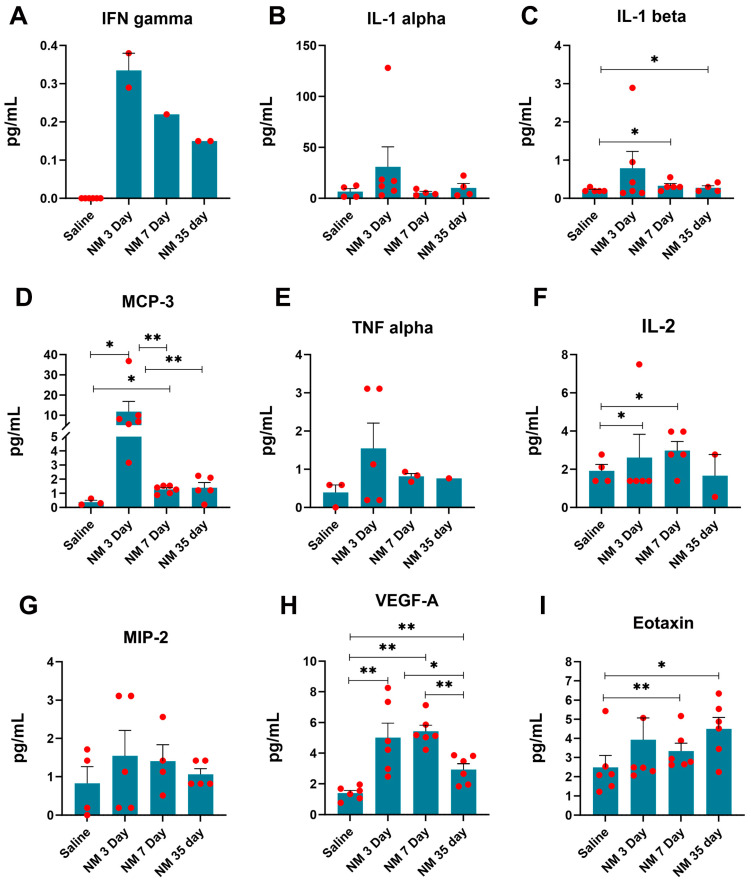
Assessment of inflammatory markers following NM exposure. The ocular surface tissue, including cornea, sclera, and conjunctiva, was micro-dissected from enucleated eyes of saline- and NM-exposed mice at 3, 7, and 35 DPI. Values represent mean ± SEM (n = 6; * *p* < 0.05, ** *p* < 0.01). All markers were not expressed in all the samples; the red dots indicate the number of samples out of six that are positive for that particular marker.

**Figure 4 ijms-25-00742-f004:**
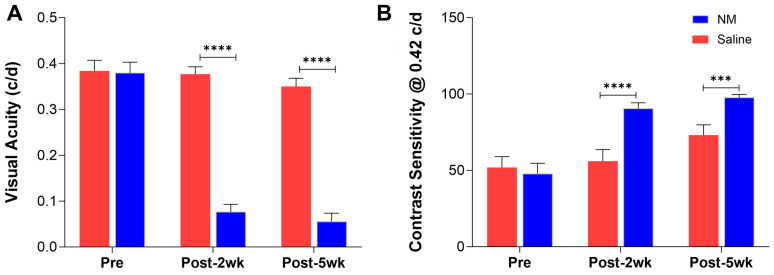
NM exposure leads to comprised vision. Visual acuity (**A**) and contrast sensitivity (**B**) were measured using OKN as described in the Section 4 of saline- and NM-exposed mice before treatment (Pre), at 14 DPI (Post-2wk), and at 35 DPI (Post 5wk). Values represent mean ± SEM (n = 6; *** *p* < 0.001, **** *p* < 0.0001).

**Figure 5 ijms-25-00742-f005:**
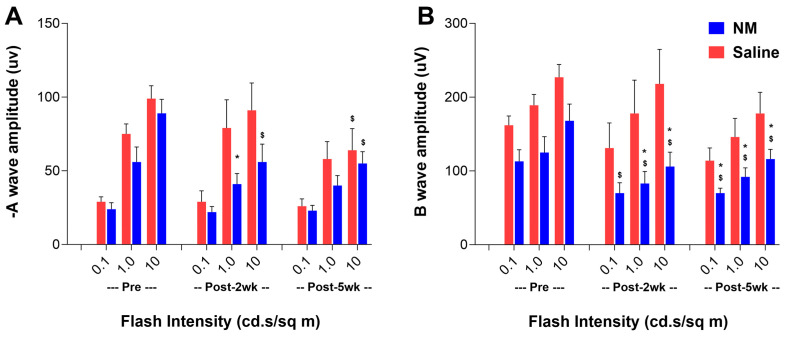
Retinal functions were affected following NM exposure. Retinal functions were evaluated by scotopic ERG as described in Section 4 of saline- and NM-treated mice before treatment (Pre), at 14 DPI (Post-2wk), and at 35 DPI (Post-5wk). Values represent mean ± SEM (n = 6; * *p* <0.05: saline vs. NM; ^$^ *p* < 0.05: NM Pre vs. Post).

**Figure 6 ijms-25-00742-f006:**
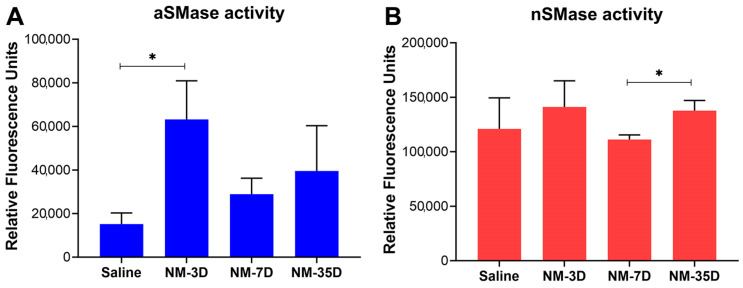
Altered activity of sphingomyelinase (SMase) enzyme following NM exposure. The activity of acidic (**A**) and neutral (**B**) SMases was measured from the ocular surface tissue of saline- and NM-treated mice on designated days post-exposure. Values represent mean ± SEM (n = 6; * *p* < 0.05).

**Figure 7 ijms-25-00742-f007:**
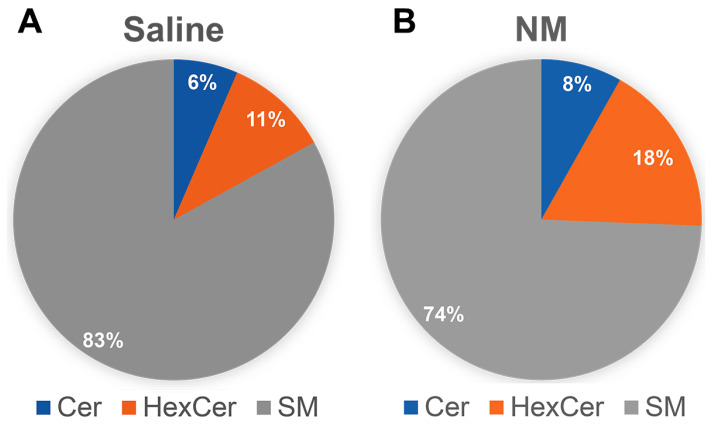
Sphingolipid homeostasis is affected by NM exposure. Ocular surface tissues were isolated from mice treated with saline and at 3 DPI of NM exposure and were analyzed for sphingolipid levels following published protocols as described in Section 4. Values represent mean ± SEM (n = 6).

**Figure 8 ijms-25-00742-f008:**
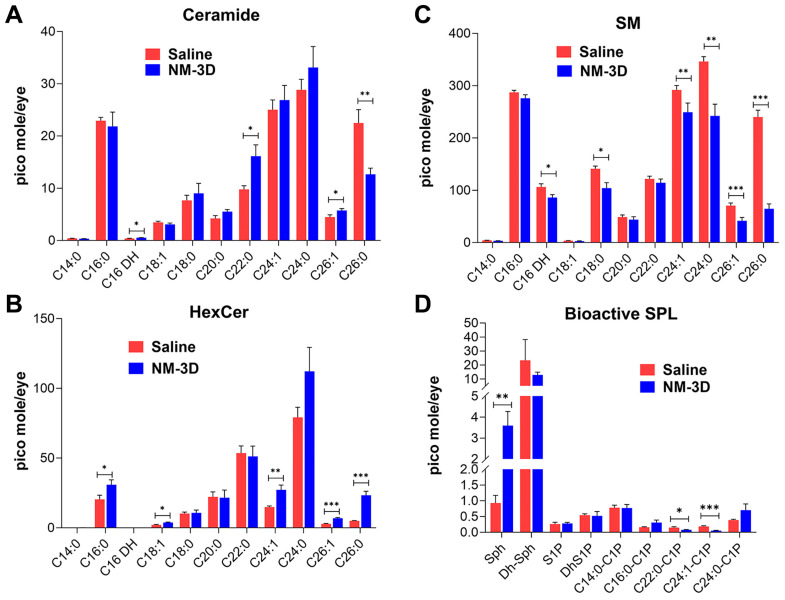
Sphingolipid species are altered due to NM exposure. Levels of different species of sphingolipids of the ocular surface tissue from saline and NM-treated mice at 3 DPI were analyzed via liquid chromatography and mass spectrometry (LC-MS/MS). Species of sphingolipids with different carbon chains [e.g., C14:0 = 14 carbon, no unsaturation; C24:1 = 24 carbon, 1 (mono) unsaturation] from the groups of ceramide (**A**), HexCer (**B**), SM (**C**), and bioactive sphingolipids (**D**) that includes sphingosine (Sph), dihydro (Dh)-Sph, sphingosine 1-phosphate (S1P), and Dh-S1P and species of ceramide 1-phosphate (C1P). Values represent mean ± SEM (n = 6; * *p* < 0.05, ** *p* < 0.01, *** *p* < 0.001).

## Data Availability

Data are contained within the article.

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
