# Peer review of "Mouse Model of Nitrogen Mustard Ocular Surface Injury Characterization and Sphingolipid Signaling"

_ijms, 2024, doi:10.3390/ijms25020742_

Round 1

Reviewer 1 Report

Comments and Suggestions for Authors

Interesting study, but poorly presented.   A lot of work has been done, but it is not present in a clear manner. 

Abstract - will change after manuscript has been rewritten

Introduction - a lovely review and explanation.  However, that is not the point of an introduction.  The point is to clearly and succinctly explain the area of interest ( with definitions explained - sort of done) explain what has been done specifically in this filed (absent) and clearly state the purpose of this study and how it add to the field (absent) 

material and methods: much missing information

At no stage did the authors indicate how many animals were used (both in total and per technique). (sections 4.1, 4.2, 4.3, 4.4, 4.5, 4.6, 4.7 and 4.8)

line 551: what was the fixative

line 551 - you mean "processed for paraffin histology"

line 551: how thick were the sections?  Was a rotary microtome used?  Which brand?

line 464: 48 antibodies?  What were controls?  What were the antibodies - only a few (I presume the successful one) ere shown in results.  Supplementary data?

line 603: branch of microplate reader?

Line 552: which H&E technique (there are many - reference)

Results: As it currently reads, it is a mishmash of methodology, results, introductory material and interpretation.

e.g.: lines 91-100 are redundant - already in material section, why repeat?

lines 101-103 - interpretation - discussion

remainder of 2.1: Not all those figures (in figure 1) are necessary --why show 21 individual eyes when you really only need about 8.  Also, need scale bars.  As for the accompanying legend - too wordy, only need to indicate what is there, not explain what is being shown,

Lines 155-156: three groups - without overlaps ( currently, day 7 is in two groups!  try: early acute (1-6 DPI), acute (7-14 DPI) and chronic (21-35+ DPI) phases.

2.2.: lines 170-173 is needed in introduction

Once again - figure 2 has too many images - choose the best 7-8.  Scale bars?  if you are referring to actual epithelium - then you need higher magnification lines 206-208 - methodology - delete.

2.3.: lines 212-216 = introductory, lines 217-221 - methodology, lines 241-254 - discussion

2.4.: line 2640267  - introductory, lines 275-276  - discussion 

line 270 - severely affected?  what statistics to support that vague statement?  NM caused decrease in visual acuity but increased in contrast sensitivity with time?  You results are more complex than the vague assertions put forth.

figure 5 - what does 0.1, 1.0 and 10 refer to in the x axis?

2.5.: lines 289-294 - introductory, lines 294-296 - methodology.

lines 311-313 - this comes out of nowhere.  Since there is no explanation of why any methods were used, why now a singular point?

lines 324-326 - data is far more complex than explained here

lines 338-342 - discussion

Discussion - well written random paragraphs about the topic, not really explaining the significance of the results.  There is a lot or reiteration of results and even introductory material. Rewrite!

This is what you need to discuss: The three stages!  Is this a novel way of examining orbital damage?  if so, then that is the focus of your discussion, using section headers.

1) Normal - significance of all your findings

2) early acute - significance of your findings (e.g.: antibodies, sphingomyelinase activity etc..)) and how it compares to other studies

3) acute - significance of your results (e.g.: antibodies, sphingomyelinase activity, vision, etc..) and how it compares to other studies

4) Chronic (and recovery without treatment?) - significance of your results (e.g.: antibodies, sphingomyelinase activity, sphingolipid profiling, vision etc..) and how it compares to other studies.

Conclusion - rewrite, shorten to one paragraph of what your found (three phases) and what needs to be done next!

No controls were mentioned in the manuscript for any of the data collected.

Comments on the Quality of English Language

It was so badly organized, I paid little attention to the language issues.  It seemed fine.

Reviewer 2 Report

Comments and Suggestions for Authors

The authors developed a model of nitrogen mustard ocular injury in mice and characterized this model by clinical and histopathological analysis of the ocular surface, quantitative analysis of the inflammatory markers and sphingolipids, and visual and retinal functions at both acute and chronic phases of the injury. They observed severe eye inflammation, leading to structural deformity of the corneal epithelium and stroma and diminished visual and retinal functions starting at the acute phase and continuing through the chronic phases. Alterations of the inflammatory markers and their expression at different phases of the injury, along with an increase in bioactive sphingolipid ceramide and a reduction in sphingomyelin levels were also seen.

The paper contains new material, and a big plus is a study of visual functions that goes beyond the ocular surface and is clinically important. The experiments seem to be well controlled, with adequate numbers of animals analyzed.

This reviewer has several concerns about the manuscript.

1. It seems incorrect to claim novelty of the mouse model. There were previous papers published by several groups, notably by Dana’s group and Djalilian’s group [both uncited; Soleimani et al. Exp Eye Res 2023; Soleimani et al. Cells 2023; An et al. Int J Mol Sci 2022; Alemi et al. Exp Eye Res 2023 (2 papers)]. Even the paper of a co-author is not cited (Mishra et al. J Pharmacol Exp Ther. 2023). The claim for novelty should be removed or significantly toned down, as the differences are minor (dose and exposure time).

2. Unlike Djalilian group’s work (An et al. 2022), the authors only used one dose and did not justify it, nor tested other doses and exposure times. The justification is needed.

3. The novelty of studying sphingolipids in NM injury is minimal.

4. It is unclear whether animals were of both sexes or of only one. Please specify. Also, for better comparison, one eye is usually treated.

Reviewer 3 Report

Comments and Suggestions for Authors

Sandip K. Basu et al. developed and characterized a murine model of NM-induced ocular surface injury. In this model, they also evaluated histological changes, markers of inflammation sphingolipid analysis, and vision function. The manuscript is well written; however, I have some remarks to make.

1.      It would be more appropriate to include the total number of animals used, and the number of animals in each experimental group.

2.      In Figure 1, please ameliorate the resolution of pictures.

3.      It would be more appropriate to place the caption of Figure 1 below the reference figure.

4.      Figure 2 is very confusing. Scale bars and magnifications are missing. From panel A to D (same as from E to I; J to N and O to S) do the magnification pictures come from the same animal in the same cornea section? If they come from different animals are not comparable and the authors should prepare the panel differently in a clearer manner.  

5.       In ERG analysis, the “a wave” is a negative wave, representing the response of photoreceptors to the light stimulus, and the “b-wave”, or positive wave, represents the response of cells deputed to light signal transmission. Amplitude, measured between the a- and b-wave peaks, is considered an index of retinal function. The amplitude of the b-wave shown in Figure 5, as written in the methods, is the variation of amplitudes (b-wave-a wave) between different experimental groups, so it is an error to name it b wave. The calculation of "wave b" is the same as that of "wave a".

6.       Line 423, please correct “regrding”.

7.       Lines 450 and 463, please correct the references' character.

8.       The discussion is too verbose and is likely to confuse the reader; I recommend shortening it.

Round 2

Reviewer 2 Report

Comments and Suggestions for Authors

The authors did a great job revising the paper. No further concerns from this reviewer. Just one suggestion: it is customary to use Greek symbols when spelling alpha, beta, etc. It is up to the authors to change it or leave, as all symbols are uniformly spelled out.